# Machine Learning-Assisted Large-Area Preparation of MoS_2_ Materials

**DOI:** 10.3390/nano13162283

**Published:** 2023-08-09

**Authors:** Jingting Wang, Mingying Lu, Yongxing Chen, Guolin Hao, Bin Liu, Pinghua Tang, Lian Yu, Lei Wen, Haining Ji

**Affiliations:** School of Physics and Optoelectronics, Xiangtan University, Xiangtan 411105, China; 202005710126@smail.xtu.edu.cn (J.W.); 202021001488@smail.xtu.edu.cn (M.L.); 202121521308@smail.xtu.edu.cn (Y.C.); guolinhao@xtu.edu.cn (G.H.); liubin@xtu.edu.cn (B.L.); pinghuatang@xtu.edu.cn (P.T.); 202005710108@smail.xtu.edu.cn (L.Y.); 202205710203@smail.xtu.edu.cn (L.W.)

**Keywords:** machine learning, Gaussian regression model, CVD, MoS_2_, area prediction

## Abstract

Molybdenum disulfide (MoS_2_) is a layered transition metal-sulfur compound semiconductor that shows promising prospects for applications in optoelectronics and integrated circuits because of its low preparation cost, good stability and excellent physicochemical, biological and mechanical properties. MoS_2_ with high quality, large size and outstanding performance can be prepared via chemical vapor deposition (CVD). However, its preparation process is complex, and the area of MoS_2_ obtained is difficult to control. Machine learning (ML), as a powerful tool, has been widely applied in materials science. Based on this, in this paper, a ML Gaussian regression model was constructed to explore the growth mechanism of MoS_2_ material prepared with the CVD method. The parameters of the regression model were evaluated by combining the four indicators of goodness of fit (r2), mean squared error (MSE), Pearson correlation coefficient (p) and *p*-value (*p*_val) of Pearson’s correlation coefficient. After comprehensive comparison, it was found that the performance of the model was optimal when the number of iterations was 15. Additionally, feature importance analysis was conducted on the growth parameters using the established model. The results showed that the carrier gas flow rate (Fr), molybdenum sulfur ratio (R) and reaction temperature (T) had a crucial impact on the CVD growth of MoS_2_ materials. The optimal model was used to predict the size of molybdenum disulfide synthesis under 185,900 experimental conditions in the simulation dataset so as to select the optimal range for the synthesis of large-size molybdenum disulfide. Furthermore, the model prediction results were verified through literature and experimental results. It was found that the relative error between the prediction results and the literature and experimental results was small. These findings provide an effective solution to the preparation of MoS_2_ materials with a reduction in the time and cost of trial and error.

## 1. Introduction

MoS_2_ is a typical layered transition metal-sulfur compound. Single-layer MoS_2_ is a sandwich structure with S-Mo-S overlapping, which has a good band gap structure [1]. Under certain external conditions, it will transit from an indirect to direct bandgap semiconductor, exhibiting strong photoluminescence when confined in a 2D monolayer [2]. It is expected to be a candidate for next-generation application in the field of nanoelectronics devices, which has aroused the interest of many researchers. As early as 2011, B. Radisavljevic and A. Radenovic et al. reported that semiconductors made of single-layer MoS_2_ thin films can be used to manufacture electronic chips with smaller volume, better performance and higher efficiency [3]. In recent years, the preparation of large-area, high-quality MoS_2_ thin films has become the focus of research, and its application areas have been constantly expanding, with typical representatives being next-generation nanoelectronics [4,5], biosensors [6], solar cells [7,8], etc. However, the area of MoS_2_ has an extremely important impact on the characteristics of the prepared devices. For example, a new single heterojunction solar cell structure based on n-type MoS_2_ and WS_2_ was prepared with its photovoltaic characteristics being greatly affected by the area of MoS_2_ and WS_2_ [9]. At present, the common method for preparing MoS_2_ is chemical vapor deposition (CVD). CVD, a method of reacting chemical gases or vapors on the surface of a matrix to synthesize coatings or nanomaterials, is the most widely used technology in the semiconductor industry for depositing a variety of materials, including a wide range of insulating materials, most metal materials and metal alloy materials. In theory, it is simple: after the introduction of two or more gaseous raw materials into a reaction chamber, they chemically react with each other to form a new material, which is deposited on the wafer surface. The single layer of MoS_2_ generated on the substrate often has a certain triangular shape, and the reaction needs to be carried out under high-temperature conditions. If the reaction time is too long, certain sulfur, molybdenum and their compounds will be deposited on the generated MoS_2_, which will have a considerable impact on the area of the generated MoS_2_. Lee et al. obtained a continuous MoS_2_ film with a size of up to 2 mm through the vulcanization of MoO_3_, but the film was not a uniform monolayer [10]. Hence, the preparation of large-area and high-quality MoS_2_ thin films remains a challenge.

With the deepening of computer science research, an increasing number of machine learning algorithms, such as support vector machines, naive Bayes, artificial neural networks, decision trees, etc., are being widely used. In chemical synthesis, James M. Tour et al. used the parameters of mass, capacitance, voltage, pretreatment and duration and then applied an ensemble of models to predict the yield of graphene, an extremely important material in wastewater treatment [11]. Chen et al. successfully achieved the two-step hydrothermal synthesis of VO_2_ under the guidance of machine learning and evaluated the performance of the machine learning model. The RF model had the best performance, with a prediction accuracy of 87.27% [12]. Chen et al. realized the controllable synthesis of multicolor discs with the assistance of machine learning, providing new ideas for the rational design and improvement of optical disc performance in the future [13]. Our research group successfully achieved controllable preparation of MoS_2_ using an extreme gradient boosting (XGBoost) model with an average prediction accuracy of over 88% and an AUC value of up to 0.91. The multilayer perceptron (MLP) model, with an average prediction accuracy of 75% and an AUC value of 0.8, was used to successfully realize the controllable preparation of MoS_2_ materials [14]. Machine learning can largely compensate for the shortcomings of cumbersome experimental steps, long experimental time and high cost in the traditional material research process. However, further exploration is needed to prepare MoS_2_ films with larger areas for better applications.

Therefore, a Gaussian regression model was constructed using machine learning via the collection of 200 sets of experimental data obtained in the laboratory and literature. Furthermore, the best model parameters were found to optimize the synthesis conditions and predict the experimental results by changing the iteration number of the model cross validation, thus providing theoretical support for the MoS_2_ preparation of a given area.

## 2. Methods

Machine learning reveals the potential relationship between synthesized feature data through training and learning models and then fits the experiments to select the optimal experimental synthesis conditions. The research of machine learning in materials mainly includes, as shown in Figure 1, data acquisition and processing, feature engineering, model building, simulation data analysis and verification [15]. MoS_2_ is prepared by adding precursors (molybdenum and sulfur sources) and controlling the macroscopic parameters of the reaction (such as reaction temperature, the ratio of the two molybdenum trioxide and sulfur element precursors, carrier gas flow rate, reaction time, etc.). For the purposes of this paper, the larger the side-length size of the synthesized triangle MoS_2_ is, the larger its area.

### 2.1. Data Acquisition and Processing

To form a dataset, 200 sets of experimental conditions for CVD synthesis of MoS_2_ were collected from the literature and laboratory (see Appendix A). The molybdenum-to-sulfur ratio, carrier gas flow rate, reaction temperature, reaction time and side-length dimensions of the synthesized triangular MoS_2_ were recorded. Due to the different selection of samples, the collected data were preprocessed to remove some duplicate experimental conditions and results. Subsequently, 200 sets of data in the dataset were statistically analyzed, with the scatterplots of each characteristic variable and experimental results being shown in Figure 2. The 200 sets of experimental condition data in our dataset originated from different laboratories, and there might have been a few subtle differences in the experimental conditions. In our study, apart from the four characteristic parameters selected, other parameters were not the main factors affecting the experimental results. Rather, the other parameters were idealized and considered consistent during the experiment process.

The triangular MoS_2_ edge lengths synthesized with CVD in the dataset ranged from 0.5 μm to 300 μm. Among them, those greater than 30 μm were defined as delimiting a large area [16,17] and accounted for 48% of the total. The area size of MoS_2_ corresponds to different experimental conditions, and there is an urgent need for a machine learning model that can predict the generation area of MoS_2_ based on the input of experimental conditions. In this study, the Gaussian regression model was used to fit the influence of each feature parameter on the experimental results to predict the area size of the obtained MoS_2_.

### 2.2. Feature Engineering

In order to reduce model computation time and make the fitting effect of the regression model better, it is necessary to select the feature data that are a completely decisive factor for the synthesized area of MoS_2_. Four characteristic parameters, including the ratio of two precursors (molybdenum trioxide and sulfur, R), carrier gas flow rate (Fr), reaction temperature (T) and reaction time (Rt) were selected during the establishment of the model. The characteristics of the dataset for describing the potential relationship between the size of MoS_2_ and the selected characteristic values are shown in Table 1; the distribution of each feature parameter data in the dataset is described through means and standard deviations. In machine learning, the Pearson correlation coefficient is a statistical data test used to reflect the degree of similarity between two variables. It can be used to determine whether the relationship between the extracted feature descriptors and categories is positive, negative or noncorrelated. The correlation among the four selected feature parameters is shown in Figure 3. Among them, the molybdenum-to-sulfur ratio was positively correlated with the reaction temperature and reaction time. In addition, the reaction time and the carrier gas velocity were also positively correlated. Among all the correlation coefficients, the maximum value was 0.21, indicating that the independence of the different variables was good and that there would be little influence exerted between them.

There are many experimental variables in the preparation of MoS_2_. In addition to the above four characteristic parameters, there were other parameters that needed to be considered in the actual CVD preparation of the MoS_2_ experiment, such as the heating rate during the reaction, potential addition of NaCl, boat configuration, and the pressure in the quartz tube. In the reaction process, NaCl can be added in order to promote the formation of a monolayer of molybdenum disulfide film and the increase of film area. Studies have shown that mixing metal oxides with salts can generate droplets or metal chloride oxides, which increase the mass flux and vapor pressure of the Mo source as well as the reaction and nucleation rate [18]. However, it does not affect the resulting molybdenum disulfide film. The boat configuration has a certain influence on the uniformity of the molybdenum disulfide film. It has been also found that the heating rate of the reaction is closely related to the pressure in the quartz boat during the reaction. If these two parameters are trained in learning, the reliability of the model will be reduced, which will affect the model evaluation index and prediction results. The environmental conditions of the laboratory, mainly those related to temperature and humidity, can also have a certain impact on the experimental results,. However, in the process of preparing molybdenum disulfide with CVD method, the whole experiment was carried out in a CVD furnace, and the synthesis of molybdenum disulfide was less affected by laboratory environmental conditions. Due to the large amount of literature exploring the influence of the four characteristic quantities on the prepared thin film area, no other characteristic parameters were selected to describe this experiment. Although these feature parameters were not selected to explore their impact on the experimental results, controlling the consistency of other variables during the experimental process could reduce the impact of the experimental results.

The molybdenum source commonly used for preparing MoS_2_ is molybdenum trioxide powder, and the sulfur source is sulfur powder. The schematic diagram of the experimental boat is shown in Figure 4. The positions of the precursor molybdenum source and sulfur source are at the center of the quartz boat, and the substrate material is SiO_2_. The carrier gas is the inert gas argon. The cooling method is natural cooling. By fixing the above parameters to reduce the parameters in the training process of the model, we could shorten the training time of machine learning.

### 2.3. Model Establishment

Gaussian process regression (GP_R) is a regression model based on the Gaussian process. This model obtains a Gaussian distribution function by fitting the data to obtain accurate prediction results [19]. Gaussian process (GP) is a commonly used stochastic process model. In the field of machine learning, GP is a machine learning method developed on the basis of the Gaussian stochastic process and Bayesian learning theory. It is strictly anchored in statistical learning theory and has a good adaptability in dealing with complex problems, such as high dimensions, small samples, nonlinearity, etc. The Gaussian regression model has greater flexibility and prediction accuracy than does traditional regression, and it can adapt to different data distributions and model complexity with more accurate prediction results than can traditional regression (linear or quadratic). Therefore, the Gaussian regression model was deemed highly suitable for the data set collected. GP has also become a research hot spot in the field of machine learning and been successfully applied in many fields [20,21,22,23,24].

There is a complex correlation between the experimental results and the selected feature parameters, but the number of datasets obtained was relatively small. Hence, cross-validation was conducted on the established regression model to verify its performance [25]. The principle of this process is to divide the dataset into two parts: a training set and a test set. The model was trained through the training set, and the performance of the model was verified using the testing set. During the model training process, GridSearchCV, only applicable to small datasets, was used to automatically adjust the relevant parameters. This method is highly suitable for our dataset although it is more time-consuming [26,27]. Cross-validation can effectively evaluate model performance and avoid overfitting and underfitting problems. In terms of model performance evaluation, four indicators were selected to obtain the conditions for the best model, including goodness of fit (r^2^), mean squared error (MSE), Pearson correlation coefficient and *p*-value of Pearson correlation coefficient [28]. Among these, goodness of fit (r^2^) and mean-squared error (MSE) were used to evaluate the model training process. Pearson correlation coefficient and *p*-value of Pearson correlation coefficient were used to determine the correlation between feature parameters. Goodness of fit reflects the extent to which the model fits the data, with values ranging from 0 to 1: the closer to 1, the better the model fits the data. r^2^ equal to 1 indicates that the model fits the data perfectly, while r^2^ equal to 0 indicates that the model is unable to explain the variance of the data. Mean squared error (MSE) reflects the gap between the predicted and actual values of the model: the smaller the value, the better the prediction performance of the model. Pearson correlation coefficient is an indicator used to measure the degree of linear correlation between two variables, with values ranging from −1 to 1: the closer to 1, the higher the positive correlation between the two variables; the closer to −1, the greater the negative correlation between the two independent variables; the closer to 0, the less linear the relationship between the two variables. The *p*-value of Pearson correlation coefficient is used to test whether the linear correlation between two variables is significant: the smaller the value, the more significant the linear correlation between the two variables.

## 3. Results and Discussion

### 3.1. Model Results and Analysis

In the established Gaussian regression model, a group of model conditions with the best evaluation index was selected to predict the area of MoS_2_. Two performance indicators under six different iterations were processed and statistically analyzed, and the mean values of each evaluation indicator were obtained, as shown in Figure 5. Usually, the increase of the number of iterations in cross-validation will continuously improve the performance of the model, but setting the value of the number of iterations too large often leads to the overfitting of the model. Research indicates that r^2^ continues to increase in iterations from 5 to 15 and then begins to decrease as the number of iterations continues to increase [29,30,31]. Moreover, when the number of iterations is 15, the MSE reaches its minimum value, indicating that the model has good prediction performance. In summary, it can be concluded that the optimal number of iterations is 15.

### 3.2. Optimization of Synthesis Conditions

Synthesis conditions play a crucial role in the preparation of molybdenum disulfide via CVD. However, in practical experiments, there are always cases where the reaction is incomplete, the reactants do not react sufficiently, or the concentration of the reactants is impure. The amount of reactants needed for the experiment is always more than the theoretical amount. Therefore, in the actual experiment, the experimenter should select the dosage ratio of the two precursors most suitable for the preparation of molybdenum disulfide with CVD according to the experimental conditions. In addition to this, the reaction temperature (T) affects the growth rate of molybdenum disulfide. The temperature affects the rate of gasification of precursor S and the reaction rate of preparing molybdenum disulfide with CVD. Therefore, the reaction temperature should be considered. If the reaction temperature is too high, a series of problems will occur. Firstly, the evaporation rate of precursor S source is accelerated, which will lead to insufficient reaction of the molybdenum source. Secondly, it can also cause excessive internal pressure in the quartz tube and require a long time during the natural cooling process. Thirdly, this will have a certain impact on the service life of the experimental device. If the reaction temperature is too low, the evaporation of the S source and the flow rate of the carrier gas will decrease, which will result in a lower growth rate of molybdenum disulfide and a longer reaction time. Therefore, whether in the preparation of molybdenum disulfide with CVD in actual experiments or the conclusion obtained through machine learning model training data, the reasonable selection of carrier gas flow, molybdenum-to-sulfur ratio and reaction temperature during the reaction process is of great significance to the formation of molybdenum disulfide. Additionally, it can be stated that machine learning plays an important role in the analysis and selection of characteristic parameters in the preparation of molybdenum disulfide with controllable area in CVD.

In order to optimize the synthesized feature parameters, the SHAP (SHapley Additive exPlanations) library in the XGBoost model was used to extract the feature importance from the four selected feature parameters. SHAP is an explanatory technique used to explain the effect of each feature parameter on the output in the model [32]. Feature importance analysis provides an estimate of the predictive power of all the features used to train an ML model. It can be seen from Figure 6 that the characteristic parameter that has the greatest impact on the film area is the flow rate of the carrier gas (Fr), followed by the molybdenum sulfur ratio (R), with the reaction time (Rt) having the smallest impact.

With reference to a large number of literatures, a large area of MoS_2_ was defined as a triangle with a size range greater than 30 µm, and further statistical analysis was conducted on the predicted area to determine the optimal experimental conditions. This model was used to predict the MoS_2_ size of a dataset composed of 185,900 virtual experimental conditions data, 159,352 of which were located in a large area, accounting for 85.72%. Based on the prediction results, the corresponding molybdenum-to-sulfur ratio, carrier gas flow rate, reaction temperature and reaction time could be optimized. Therefore, the preparation conditions for large area MoS_2_ were be optimized, as shown in Table 2.

### 3.3. Model Verification

The area of MoS_2_ predicted with machine learning might have contained a degree of deviation, so the model needed to be verified. Firstly, four sets of experimental conditions and results that were not used for model training were found from different literature sources. The model was validated using the prediction results of the model under the corresponding conditions, as shown in Table 3. The results from the literature were compared with the predicted results of the model. It was found that the relative errors were 3.19%, 4.40%, 0.68% and 0.67%, respectively. In addition, the experiments were conducted to further verify the reliability of the model. The SEM images of MoS_2_ prepared under experimental conditions of R = 0.128, Fr = 175 sccm, T = 800 ℃ and Rt = 30 min, with a measured edge length of 43.571 μm, are shown in Figure 7a. Under this condition, the predicted edge length of the model was 42.302 μm. The relative error between the experiment and prediction was 2.91%. The SEM images of MoS_2_ prepared under experimental conditions of R = 0.01, Fr = 100 sccm, T = 800 ℃ and Rt = 15 min, with a measured edge length of 35.869 μm, are shown in Figure 7b. The predicted edge length of the model was 37.975 μm. The relative error was 5.87%. In summary, both literature validation and experimental validation showed that the relative error did not exceed 6%, indicating high reliability of the predicted results. These results prove that the trained machine learning model can assist in the preparation of large-area molybdenum disulfide. To a certain extent, it provides practical guidance for the growth of molybdenum disulfide in combination with different growth parameters.

## 4. Conclusions

In this study, machine learning was used to examine the preparation of large-area MoS_2_ materials via the Gaussian regression model. First, the model was applied to extract the feature importance in the dataset. Then, under the optimal model with 15 iterations, size prediction was performed on the simulated data that included 189,000 data sets. Furthermore, the growth conditions were optimized, and the optimal range of each growth condition was obtained. Finally, the model was validated. The results showed that the model could be used to predict the area obtained under experimental conditions in the literature, and the maximum difference between the literature results and the predicted results with a max relative error was 4.40%. Meanwhile, experimental verification was conducted on the predicted area of the model, with a max relative error of 5.87%. These results also indicate that machine learning has significant advantages in assisting in the preparation of large-area MoS_2_ materials and provide an important foundation for the subsequent preparation of large area two-dimensional materials.

## Figures and Tables

**Figure 1 nanomaterials-13-02283-f001:**
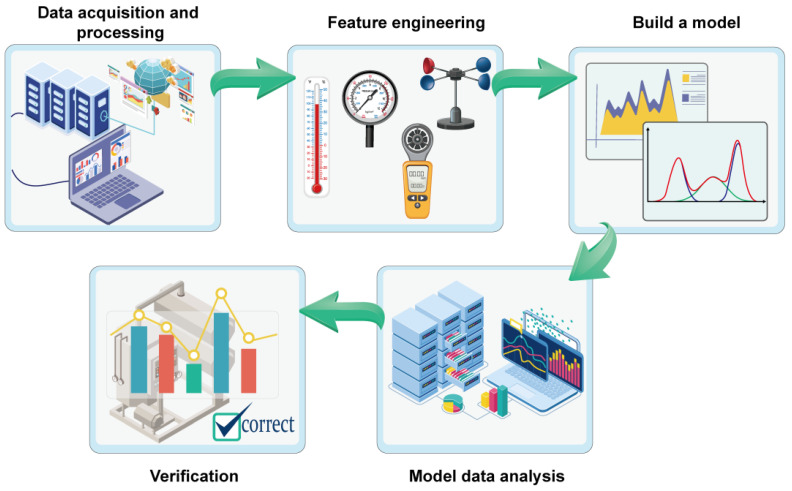
Machine learning-assisted CVD synthesis of a large-area MoS_2_ workflow diagram.

**Figure 2 nanomaterials-13-02283-f002:**
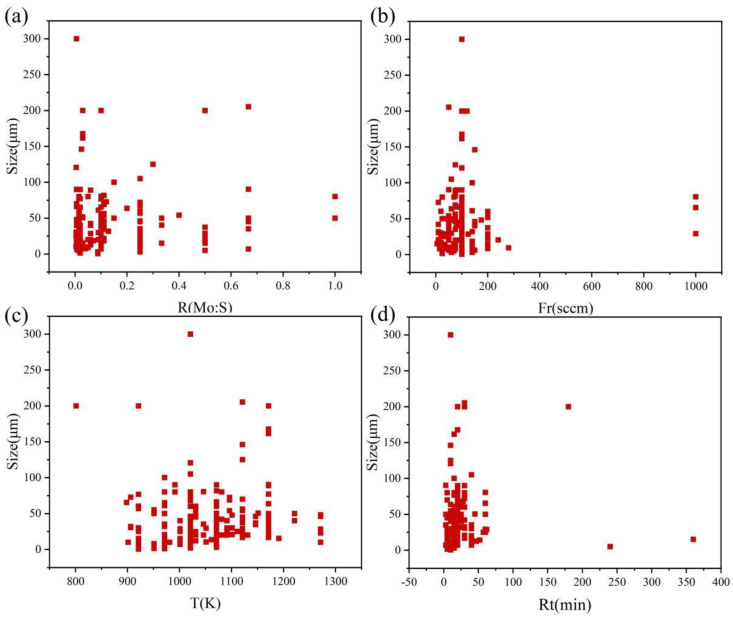
Scatter plots of each feature parameter and size in the dataset. (**a**) Molybdenum sulfur ratio, (**b**) carrier gas flow rate, (**c**) reaction temperature, (**d**) reaction time.

**Figure 3 nanomaterials-13-02283-f003:**
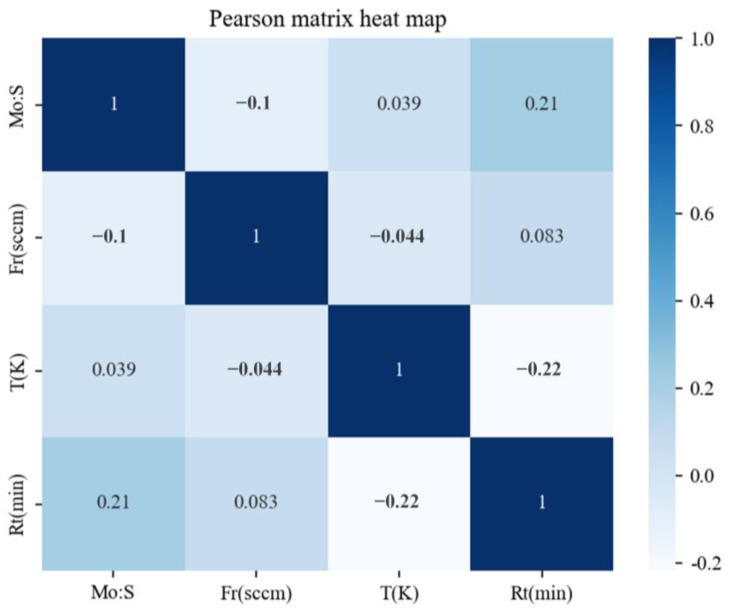
Pearson matrix heat map of the four characteristic parameters in CVD synthesis of MoS_2_.

**Figure 4 nanomaterials-13-02283-f004:**
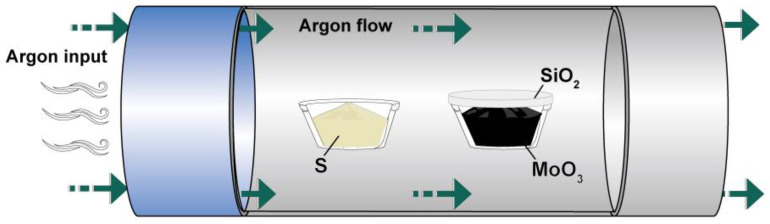
Schematic diagram of the MoS_2_ experimental vessel prepared via CVD.

**Figure 5 nanomaterials-13-02283-f005:**
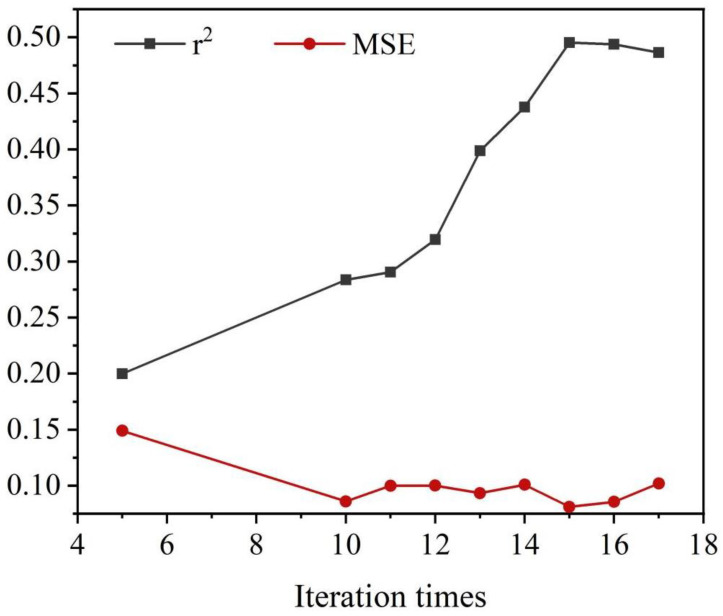
The curves of the evaluation indicators (r^2^, MSE) in the Gaussian regression model with increasing iterations.

**Figure 6 nanomaterials-13-02283-f006:**
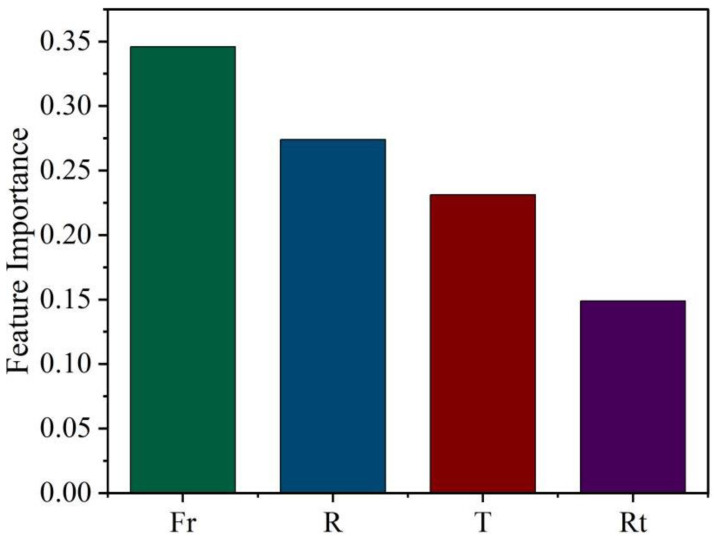
The importance of the features extracted from the SHAP library, which was learned from a sample of 200 datasets, with Fr and R being the two most important features.

**Figure 7 nanomaterials-13-02283-f007:**
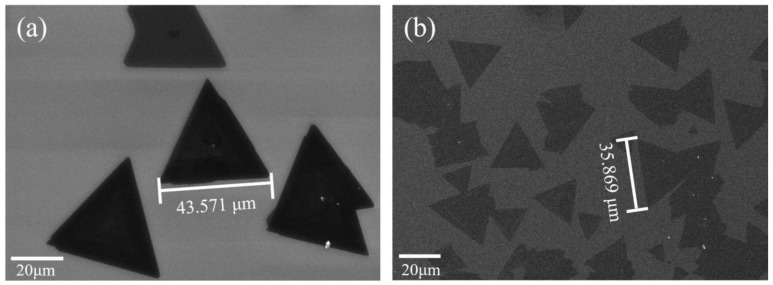
SEM images of MoS_2_ prepared with CVD under different experimental conditions. (**a**) R = 0.128, Fr = 175 sccm, T = 800 ℃, Rt = 30 min. (**b**) R = 0.01, Fr = 100 sccm, T = 800 ℃, Rt = 15 min.

**Table 1 nanomaterials-13-02283-t001:** Characteristics of the MoS_2_ dataset prepared with CVD.

Notation	Feature	Unit	Mean	Standard Deviation
R(Mo:S)	Molybdenum-to-sulfur ratio		0.12	0.18
Fr	Gas flow rate	sccm	105.10	120.70
T	Reaction temperature	K	1045.36	82.41
Rt	Reaction time	min	22.39	33.15

**Table 2 nanomaterials-13-02283-t002:** Optimization range of growth conditions for MoS_2_ generated with CVD.

Growth Parameter	Min	Max
Molybdenum-to-sulfur ratio, R (Mo:S)	0.02	1
Gas flow rate, Fr (sccm)	25	200
Reaction temperature, T (K)	850	1200
Reaction time, Rt (min)	8	75

**Table 3 nanomaterials-13-02283-t003:** Validation of the model with experimental conditions and results from the literature.

No.	R(Mo:S)	F_r_(sccm)	T(K)	Rt (min)	Size from Literature (μm)	Predicted Size (μm)	Literature Source
1	0.05	50	1098.15	10	40.639	41.935	Senkić A et al. [33]
2	0.50	200	1123.15	15	39.252	40.980	Saenz G A L et al. [34]
3	0.02	10	923.15	10	37.975	38.235	Zhang X et al. [35]
4	0.02	50	1073.15	15	25.104	24.935	Yang S Y et al. [36]

## Data Availability

Data available on request.

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
