# Peer review of "Machine Learning-Assisted Large-Area Preparation of MoS2 Materials"

_nanomaterials, 2023, doi:10.3390/nano13162283_

Round 1

Reviewer 1 Report

The manuscript “Machine Learning Assisted Large Area Preparation of MoS2 Materials” presents interesting results on the possible automation of the CVD growth parameter of molybdenum disulphide.

 The manuscript is well written and it presents interesting approach in the automation in the optimization of the growth parameters, however the paper presents one main drawback about the feedback of the CVD process that is provided by the authors. In fact, the authors consider only the size of triangular domains, without considering the thickness homogeneity and the number of layer of the MoS2 domains.  This is important because the SEM micrographs of Fig. 7 present a faint contrast, suggesting that the triangular domains can be composed of areas with different thicknesses. Can the authors give this additional feedcack to the audience?

Reviewer 2 Report

The paper reports on an attempt to utilize machine learning algorithms in the fabrication of MoS2 layers on SiO2 substrates. The main focus is on predicting an optimum combination of process parameters that leads to maximum surface areas. The topic is of interest and the given rationale, i.e., reduction of the number of "trial and error" experiments, is valid. The manuscript is logically organized and reasonably well written. Before recommending acceptance of the paper, however, I would like the authors to address the following questions and comments: 

1. Line 30: What does it mean that single-layer MoS2 has a "good band gap structure"? Good in what sense?

2. Line 45: Why are "certain sulfur, molybdenum and their compounds" deposited at long deposition times? Is the MoS2 deposition process self-limiting? If so, this needs to be stated.

3. Line 47: What does it mean that one obtains a "continuous MoS2 film with a size of up to 2 mm"? Should this be 2 mm2?

4. Line 86: What is the "fitting effect" of the regression model?

5. Lines 93/94: How does Fig. 3 show that "the independence of each variable is good"?

6. Line 97: Why would one want to add NaCl to the reaction?

7. Line 102: What does it mean that "controlling the consistency of other variables during the experimental process can reduce the impact of experimental results...". Impact on what? Aren't the experimental results the ultimate test?

8. Lines 103 to 109, and Fig. 4: How are Mo and S generated from the precursors? Why is there SiO2 in the precursor set-up? Isn't the substrate a wafer or something similar?

9. Line 146 and Fig. 5: What are the uncertainties of the obtained values? Both the peaks and valleys are rather shallow. How robust is the choice of 15 iterations?

10. Tables 1 and 3: The given numerical values show too many significant figures. For example, how does one measure a sample size of 40.639 um, or what does it mean that the mean gas flow rate is 105.05 +/- 120.73 sccm?

Acceptable.

Reviewer 3 Report

I have got serious doubts about publishing the provided manuscript in the present form.

Briefly: as a reader, I get nothing from this report. The only thing I have learnt from this text is that a Chinese team has built a model to predict the size of the MoS2 particles basing on the process parameters - but this is all. I have learned nothing about how the model was built, why the particular model was chosen, what model was obtained, etc. The reported study is not reproducible, since the authors shared no input dataset, no model development details, and no final model parameters.

Unfortunately, I have very little to write about particular drawbacks of the text, since it is too (I would say, unprecedentedly) short. Here is a list of things which, I believe, should be included by the authors to further consider this manuscript scientific content.

1. It is stated that "Our research group has used machine learning algorithms to build a model and explored the growth mechanism of MoS2 materials, successfully achieving controllable preparation of MoS210. However, further exploration is needed for the preparation of MoS2 thin films with larger areas.

Which other ML approaches have been used? What results have been obtained? Which issues have remained unexplored which demanded another model?

2. Further, it is stated that "Based on this, a Gaussian regression model was constructed using machine learning by collecting 200 sets of experimental data obtained in the laboratory and literatures."

It is not clear why this model has been chosen. Moreover, it is not quite clear why a ML approach was demanded to analyse a relatively small dataset. Why not use a conventional regression (linear or quadratic)? What is the benefit of the Gaussian regression model over simple regression, and what are its advantages over other ML instruments?

3. What is a "dataset" - 200 of which have been considered? How many observations in total were there in the complete dataset analysed by the authors? It would be ideal to share the dataset for other researches to be able to reproduce the result - or at least provide the list of references where the published data were extracted from, if the authors do not want to share their internal results publicly.

4. How exactly the model was trained? Have the authors used a commercial software to do so (if yes - which one?), or a freeware tool (which), or a own routine (which language and libraries were used?)? Which hardware was used for the training and how long did it take? Which hyperparameters were tuned or chosen during the training? How the dataset was divided into training and test subsets? How the process variables distributions different in the training and test subsets?

5. Did the authors indicate any possible outliers in the considered dataset (single observations strongly affecting the model)?

6. What are the model coefficients?

7. I did not understand the stage of the model verification at all. Given a starting dataset with at least 200 observations (I still cannot understand the dataset size) and having performed cross-validation, what is the point of verification with just 6 (4 published and 2 own) additional experiments?

8. The purpose of the model is to elucidated the conditions for the preparation of the largest crystals. What are the optimised conditions as per the model? Did the authors verify those conditions experimentally?

It is a serious concern, in my view, since the dataset visualised in Fig. 2 provides several experiments affording size > 150 μm (even up to 300 μm), but the six "verification" observations are for the size of 25-45 μm. This is the middle of the range of the response in the dataset, and it can be expected that the prediction error should be low. However, to meet the declared task, the prediction in the range of the largest size should be considered.

After providing the above-listed additional data, the manuscript should be reconsidered.

Reviewer 4 Report

In the proposed paper the authors have analysed literature parameters used in the fabrication of CVD MoS2, by means of Machine learning technique to extract the optimal fabrication parameter used. The model was confirmed by literature data and experimental preparation of samples.

·        Overall, it is not clear the main scope of the paper. It seems that the author are looking for optimal experimental parameter for fabrication of MoS2 flakes. However experimental procedures and recipes may vary from laboratory to laboratory. And fabrication parameters may depend on other experimental details such as dimension of furnace, environmental condition of the lab, humidity etc.. Can the author comment on that?

·        In the data acquisition procedure, there is no indication of the large experimental dataset indicated in the text (“a Gaussian regression model was constructed using machine learning by collecting 200 sets of experimental …” line 57). The author must indicate (maybe in the SI) the experimental works from which the data have been collected.

·        In the data acquisition the precursors are considered equal while variation of their chemical composition and quality may occur from dataset to dataset. Can the author comment on that?

·        The outcome of the model is the size of the CVD grown flakes. However, size does not represent the quality of the materials. It does not consider the occurrence of defects and the thickness (number of layers) that may be increase as effect of height and not only lateral growth.

·        It is not clear how Feature Engineering works. Is it a selection of data among the large datasets?

·        In figure 3, Pearson heat matrix must be shown with improved resolution. It seems that all the values on the diagonal are equal to 1. The author must replace the colour with the actual numeric value.

Moderate editing of English language required

Round 2

Reviewer 2 Report

The authors have addressed most of my comments in a satisfactory manner. However, there still are two questions that in my opinion, require further revision of the manuscript:

1. Comment 6: The authors' response requires more detail. How does "the addition of NaCl promote the formation of a monolayer of MoS2 film and the increase of film area"? Do the Na and Cl atoms/ions incorporate in the film?

2. Comment 10: The authors' response is highly unsatisfactory as the number of significant figures has not been addressed at all. I challenge the authors to explain how they would obtain a value of 40.639 um in an SEM and what the accuracy would be of such a measurement. The same applies to a flow rate of 105.05 sccm - an instrument that regulates flow rates at the 100 sccm level is not accurate to 0.05 sccm levels. Finally, a statement of 105.05 +/- 120.73 sccm is simply scientifically meaningless.

Reviewer 3 Report

I will start with a general comment on my final recommendation. My previous review on this manuscript stated 'major revision' although to be honest it was closer to 'reject and resubmit'. However, I see that the authors have significantly improved the text, so probably it was proper to not reject it then. This time I am stating 'minor revision' - but please be careful. Pair of the issues reported below are indeed quite serious (I mainly mean 1.1 and 1.2 in the below list). By not putting 'major revision' here I just want to show that my further assistance as a reviewer is useless. The comments are clear and straightforward enough - and therefore I can leave up to authors whether to accept them or not, and the Editor can then suggest whether the issues are indeed so important as they seem to me.

To repeat, the authors have done a great job addressing my earlier comments and making proper corrections to the text. I still wonder why these points were not covered initially, but... Below are several comments on the revised version.

1. Although the authors have put somewhat different explanations in the replies to my earlier comments, it is still stated in the Abstract that "The optimal model was used to predict the size of molybdenum disulfide synthesis under 185900 experimental conditions in the simulation dataset, so as to select the optimal range for the synthesis of large-size molybdenum disulfide." Furthermore, it is mentioned in the Introduction (lines 58-60 and 78-79). Finally, the Conclusion section states that "These results also indicate that machine learning has significant advantages in assisting the preparation of large area MoS2 materials, which provides an important foundation for the subsequent preparation of large area two-dimensional materials."

Here I have got two separate comments. 

1.1) I have failed to find the optimised conditions for the preparation of large particles. The very last sentence regarding the model states that "Using this model to predict MoS2 size of a dataset composed of 185900 virtual experimental conditions data, 159352 of which were located in a large area, accounting for 85.72%. Therefore, the preparation conditions for large area MoS2 can be optimised, as shown in Table 2." The conditions in Table 2 just report the ranges of the conditions under optimisation, not the optimised ranges. There is nothing stated in the text regarding the regions of the synthesis conditions affording the largest particles in the simulation. The verification (section 3.3) was also performed for the intermediate size of the produced particles. Overall, this means that the content of the experiment and its results contradict the stated problem and the conclusion. Either the model verification should be performed using the examples of the largest particles reported in the literature (however, I doubt that the authors will go for in) or the relevant parts of the Abstract and Conclusion should be modified accordingly.

1.2) The second issue is about the concluded advantage of the ML approach to optimise the synthesis conditions. In fact, this does not follow from the content of the reported research. The conclusion as made by the authors demands building a conventional polynomial regression using the same dataset and comparison of the MSE for the Gaussian and polynomial regressions. Instead, the results reported by the authors just show that ML can be used - but not that this approach is advantageous.

2. In line 62 - I think 'naive Bayes' should be instead of Neif.

3. In line 100 it is stated that "the collected data was pre-processed to remove some duplicate experimental conditions". I do not think this is a fully correct step (or at least not fully correct description). The presence of independent observations under identical results is indeed important, as it states the range of the output variation due to random errors or the factors not included in the model. Moreover, it is not clear how the observations to be removed were selected among the set of duplicated conditions. I believe that this sentence should be reshaped to sound more accurately.

4. In lines 174-190 it is stated that the Pearson correlation coefficient was used to assess the model training process. However (this is correctly stated by the authors, by the way), this metric catches only the linear correlation. At the same time, the authors have claimed that a strength of the Gaussian regression is the ability to 'adapt to different model complexity' - so it probably can go beyond a linear model, I guess. How appropriate is the use of the Pearson coefficient to judge about the process of training of possibly nonlinear model?

Reviewer 4 Report

The authors heve addressed the needed revisions

minor
